# Going Viral—RSV as the Neglected Adult Respiratory Virus

**DOI:** 10.3390/pathogens11111324

**Published:** 2022-11-11

**Authors:** Bethany Busack, Andrew F. Shorr

**Affiliations:** 1Department of Medicine, Medstar Washington Hospital Center, Washington, DC 20010, USA; 2Pulmonary and Critical Care Medicine Section, Medstar Washington Hospital Center, Washington, DC 20010, USA

**Keywords:** Respiratory Syncytial Virus (RSV), pneumonia, acute respiratory infection, hospitalization, influenza-like illness, elderly, burden, underestimated

## Abstract

Human respiratory syncytial virus (RSV) is increasingly recognized as a significant viral pathogen in adults with acute respiratory illness, particularly in the elderly, the immunocompromised, and those with underlying cardiopulmonary disease. Although long acknowledged as one of the most common causes of upper respiratory tract infections (URI) in children since its discovery in 1956, the true burden of disease in adults is likely significantly under-recognized. The emerging evidence of RSV as a driver of morbidity and mortality in elderly and immunocompromised patients has sparked advances in vaccine development and renewed interest in quantifying the true burden of disease. This review attempts to summarize the findings of the most recent evidence investigating the burden of RSV related disease in adults and to highlight where future research is needed.

## 1. Introduction

RSV represents one of the most important viral pathogens causing respiratory illness. Clinical symptoms vary widely, from cold-like symptoms to life threatening lower respiratory tract infections. RSV has also been implicated as a cause of exacerbations of underlying cardiopulmonary disease. Patients living in long-term care facilities, those with underlying lung and/or heart disease, the elderly, and immunocompromised patients (especially patients with hematologic malignancies actively undergoing chemotherapy) all face an increased risk for severe disease from RSV [1]. To date, insensitive RSV detection methods and the general failure to search for RSV has led many to underestimate the burden of RSV disease. In fact, a growing body of evidence shows that RSV is both more prevalent and costly than previously thought. A recent metanalysis, for example, found RSV detection to be 1.4–2 times higher when a nasopharyngeal RT-PCR test was combined with sputum RT-PCR or via paired serology specimens. Applying insights from these prior analyses to the 2022 US Census population suggests that RSV is responsible for roughly 159,000 hospitalizations, 119,000 ED admissions, and 1.4 million outpatient visits annually among US adults age ≥65 years [2].

Traditionally, RSV is often considered a less severe illness than influenza, but increasing evidence contradicts this belief. Falsey et al., for instance, completed a prospective, global study on acute respiratory tract infections in adults and observed that outcomes for patients hospitalized with RSV were worse than patients hospitalized with influenza or human metapneumovirus (hMPV) [3]. The study also found that patients hospitalized with RSV were more likely (1) to require supplemental oxygen, (2) to suffer in-hospital complications, (3) and to stay longer in the hospital. More importantly, RSV infection more frequently led to ICU admission than influenza [3]. A separate report comparing RSV and influenza in elderly adults admitted with cardiopulmonary disease found 18% of RSV patients required intensive care. Of the patients with RSV and influenza, 10% and 6% died, respectively [4,5]. Indeed, RSV places a significant burden on the US healthcare system, with annual direct costs for hospitalization exceeding $1 billion [6]. As recognition of RSV-associated illness continues to grow, so toodoes an appreciation of its importance. With viable vaccine options for the adult population in development, it has never been more critical to expand understanding and appreciation of RSV disease in adults and means for preventing it. To accomplish this objective, clinician’s require insights into both the clinical presentation of and diagnostic methods for RSV infection. Similarly, physicians must become familiar with variables that identify patients at risk for severe RSV infection and ongoing efforts at vaccine development.

## 2. Clinical Presentation

RSV causes a respiratory infection that is clinically indistinguishable from other respiratory viruses. Most commonly, patients develop upper respiratory infection (URI) symptoms such as nasal congestion and rhinorrhea (22–78%) approximately 3–5 days after initial infection [1]. In children, fever is common, and 40% of infants develop lower respiratory tract infection with cough and wheezing, clinically known as bronchiolitis [7]. The burden of RSV-related disease in infants and young children has been well characterized and will not be further elucidated in this review. In young, healthy adults, repeat infections are generally confined to the upper respiratory tract with nasal congestion, fatigue, fever, and cough being the predominant symptoms [1,8]. However, in a subset of patients the virus progresses to infect the lower respiratory tract leading to cough, wheezing, and dyspnea. A retrospective cohort study conducted in Hong Kong revealed that 70% of adults admitted to the hospital with RSV had lower respiratory tract infection complications such as pneumonia, bronchitis, or exacerbations of COPD or asthma [1,9]. Radiographically, 49.3% of patients present with imaging consistent with acute pneumonia, most commonly in the form of consolidations (23.8%) and ground-glass opacities (19.9%) that are indistinguishable from other causes of infectious pneumonia [9]. Furthermore, bacterial co-infections are common, –approximately 12.5% of patients have bacterial co-infection at presentation [1,9].

RSV is highly contagious as infection is universal in early childhood. Nearly all children are infected within the first two years of life and acquire some degree of immunity. In the Houston family study, virtually all children followed from birth were infected with RSV by two years of age, and about one half had experienced repeat infection [10]. That being said, immunity is still imperfect, and reinfections occur commonly in both children and adults [11]. Because of this, quantifying the risk of transmission is difficult. It varies greatly depending on both the setting and the population. For example, a systematic review of RSV transmission risk found that transmission risk varied from 6–56% in neonatal/pediatric settings, 6–12% in units housing immunocompromised adults, and 30–32% in other adult care settings [12].

## 3. Diagnosis

Although it is possible to detect RSV by serology, cell culture, enzyme immunoassay (rapid antigen detection test), and real time polymerase chain reaction, most methods are cumbersome and insensitive. Serology (IgM and IgG) is not useful as adult RSV infections are all reinfections. Therefore, two serum samples must be obtained demonstrating a 4-fold increase in serum antibody in order to identify acute infection [13]. Diagnosing RSV by culture is not ideal given the lability of the virus, the length of time needed for definitive diagnosis (days to weeks), and the insensitivity of diagnosis (17–39%) [1,13]. Finally, rapid antigen tests are generally not recommended for accurate diagnosis in adults as sensitivity is generally less than 10% [13]. Reverse transcription polymerase chain reaction (RT-PCR) is the gold standard method for detecting acute RSV infection with sensitivities ranging from 84–100% [1]. Collection of an adequate sample is paramount, with nasopharyngeal swabs proving more sensitive than oropharyngeal swab specimens [14]. Lower respiratory samples are preferred for intubated patients, as viral replication is greater in the lower respiratory tract in later stages of illness [1]. Regardless of the method, accurate diagnosis of RSV-related disease in adults remains imperfect. Maintaining a high index of suspicion in addition to utilizing multiple methods or sample sites is necessary for accurate identification. It is important to comment that many clinicians are not familiar with the diagnostic options for RSV thus opting not to search for RSV as a potential etiology of their patient’s symptoms since, at present, there are few practical treatment options.

## 4. Risk Factors for Severe Disease

Immunocompromised patients, patients with underlying lung and/or heart disease, the elderly, and patients living in long-term care facilities face an increased risk both for developing symptomatic RSV infection and for developing severe RSV-related disease [1]. Falsey et al., completed a systematic review of RSV-related acute respiratory infections (RSV-ARI) studies and concluded that the “likelihood of experiencing RSV-ARI for those with any comorbidity compared to those without was estimated to be 4.1” [15]. Furthermore, a prospective cohort study examining the incidence of RSV found that RSV positivity was present in “10.6% of hospitalizations with pneumonia, 11.4% of hospitalizations with chronic obstructive pulmonary disease (COPD), 5.4% of hospitalizations with congestive heart failure (CHF), and 7.2% of hospitalizations with asthma in adults ≥65 years” [6]. These observations underscore the extensive burden of this illness in patients with underlying medical conditions.

Furthermore, many now recognize the significant contribution of underlying cardiac disease in the evolution of severe RSV illness. A multicenter study of hospitalized patients with influenza-like illness in France, for instance, found that 45% of subjects with RSV also had a history of underlying heart disease [11,16]. Similarly, two prospective studies of adults with acute respiratory illness in Tennessee determined that half of patients with RSV had chronic cardiovascular disease [11,17]. In addition to just a nexus between underling cardiac disease and severe RSV infection, data now demonstrate an association between RSV positivity and exacerbations of heart failure [11]. In the cohort study described previously, it was estimated that 5.4% of all hospital admissions for CHF in the fall and winter were attributable to RSV infection [6,11].

These observations have led to hypotheses that RSV infection may directly cause myocardial injury. A case–control study assessing the link between RSV and myocardial infarction (MI) by Barnes et al. showed that patients with MIs were more likely than control subjects to have positive RSV serum IgG antibodies [11,18]. Additionally, a retrospective cohort study revealed that 16.7% of the patients with RSV who died within 60 days of admission died from an acute cardiovascular event [9,11]. The mechanism for the relationship between RSV infection and myocardial events remains unclear; some postulate that RSV infection likely causes plaque destabilization via recurrent inflammatory responses as is described with other acute viral infections [11]. Additional proposed mechanisms include a virally mediated induction of a hypercoagulable state, activation of the sympathetic nervous system leading to demand ischemia, direct RSV penetration of myocardial tissue, and transient pulmonary hypertension from the infection’s effects on the respiratory tract [11]. Regardless of the cause, given the prevalence of cardiovascular disease in the United States, understanding the relationship of cardiovascular complications in RSV positive patients is crucial [11,19]. 

With respect to mortality in the elderly, formal estimates indicate that RSV results in over 10,000 annual deaths among persons >64 years of age in the US [20]. Recent studies indicate that there are indeed twice as many RSV infections as influenza infections in older adults in the community. Length of hospital stay, need for intensive care, and mortality rates are also similar for RSV and influenza A for elderly patients [6,21]. Furthermore, “21% of older adults hospitalized with RSV require ventilatory support” [22]. Reflecting this observation, analyses of the prevalence of RSV in long term care facilities suggest an even more impressive burden of illness. In long-term care facilities, reports indicate that 5% to 27% of respiratory infections are caused by RSV. Approximately 10% to 20% of these infections cause a clinically significant pneumonia, and the mortality rate is estimated at 2% to 5% [4,20]. 

Unsurprisingly, immunocompromised patients face a greater risk of severe RSV disease and increased mortality. Moyes et al. reported that the hospital admission rate for RSV associated acute respiratory infection (RSV-ARI) was around 5–24 times more for HIV-infected older adults than for persons without HIV infection [22,23]. RSV is particularly deadly in hematopoietic stem cell translant (HSCT) patients, with progression from upper to lower respiratory tract infection occurring in 40–60% of cases. RSV lower respiratory tract infection (LRTI) in HSCT patients is associated with mortality rates exceeding 80% [1,21]. Likewise, several retrospective and one prospective study of lung transplant recipients underscore the increased mortality rates seen in this population-with mortality rates ranging from 10–20% [1,21]. Part of this mortality burden relates to RSV LRTI increasing the chance for chronic rejection (bronchiolitis obliterans syndrome/chronic lung allograft dysfunction) [1]. Studies of RSV progression in other solid organ transplants are limited to case reports. Further studies are needed in order to characterize risk of RSV-related mortality in this population [24].

The presence of underlying chronic lung disease, similar to cardiac disease, also raises the risk for severe disease requiring hospitalization. Chronic lung disease has also been found to be an independent predictor of needing ventilatory support [9]. This seems to be even more true in symptomatic RSV infections than in influenza infections. Lee et al. found that “adults admitted to the hospital and infected with RSV more often have underlying chronic lung diseases (35.6% vs. 24.1%) than do patients infected with influenza” [9]. In addition, RSV is a major cause of exacerbations in patients with asthma and COPD [1]. A post hoc analysis on COPD patients with confirmed RSV found that 48% of patients sought medical attention and 17% were hospitalized. The most common presenting symptoms in this cohort were consistent with acute exacerbation of COPD including cough (83%), increased sputum production and dyspnea (62%), and wheezing (48%) [25].

## 5. Virology and Vaccine Development

RSV is a non-segmented negative sense RNA virus in the family *Pneumoviridae*, genus *Orthopneumoviridae*. Its pleomorphic, enveloped virion ranges from spherical to rod-shaped, 120–300 nanometers in diameter [1]. RSV has two major antigenic subgroups (A and B). Although these two subgroups have antigenic differences in a number of proteins, the major genetic diversity lies within G, the attachment glycoprotein [1,4,7]. Once attached, the F (fusion) protein undergoes structural changes that cause the virion membrane to fuse with the cell membrane, initiating infection. When the F protein produced in infected cell lines reaches the surface, it causes infected cells to fuse with adjacent uninfected cells, forming “syncytia” or giant cells [4]. For several reasons the fusion protein has become paramount in the development of RSV vaccines. It is relatively conserved among strains, such that a vaccine utilizing a recombinant form of the RSV-F protein would likely be effective against both RSV A and B subgroup viruses. In addition, the “prefusion” form of the F protein has an antigenic site (theta) that induces very potent RSV neutralizing antibodies [4,26].

Currently, one passive vaccine is available, palivizumab (*Synergis*), licensed for use in high-risk infants [26]. It prevents roughly 50% of RSV related hospitalizations in premature infants, however, it is costly and must be administered monthly over the infant’s first 5-month RSV season. Palivizumab is a humanized monoclonal antibody directed against the “postfusion” form of the RSV F protein. Other monoclonal antibodies are currently in development that recognize the prefusion F protein and therefore are more effective at neutralizing RSV. These antibodies are also stabilized so that multiple doses are not necessary. 

Trials for two monoclonal antibodies that target RSV-F are currently underway, both of which have a longer half-life than palivizumab. One, developed by AstraZeneca and licensed to Sanofi, is nirsevimab (also known as MEDI8897). The other is MK-1654 (Merck). Unfortunately, both are currently being investigated only for use in the infant population [26]. A more promising development for older patients is the successful engineering of a stabilized pre-fusion conformation of RSV-F [26,27]. Several subunit vaccines utilizing recombinant RSV-F are currently undergoing clinical trials, and unlike passive immunizations these subunit vaccines target a wider population. GlaxoSmithKline recently revealed that the open label, phase 2B study of their over-60s vaccine, RSVPreF3 OA (RSV pre-F adjuvanted with AS03) resulted in significant efficacy. Furthermore, Pfizer is currently conducting a phase 3 trial of a pre-F sub-unit vaccine for adults 60 years and older (ClinicalTrials.gov: NCT05035212), with study completion estimated for June 2023 [26]. Bavarian Nordic and Jansen, in addition, are also conducting trials on potential RSV vaccines for adults.

## 6. Conclusions

Clinicians often overlook the contribution of RSV to morbidity and mortality in adults. Excluding the pediatric population, RSV is linked to roughly 250,000 deaths annually [3,28]. Though researchers and epidemiologists have made considerable progress in understanding the burden of RSV-related disease in adults, much remains yet to be discovered. For instance, data regarding RSV in developing countries is lacking. No studies exist on the incidence of RSV in any developing country [15]. This is particularly problematic when preparing global estimates of disease burden because the implications of LRIs are generally higher in older adults in developing countries. Dependence on solid fuels, malnourishment, immune-impairment, and lower socioeconomic status of the populations are all contributors to this disparity [29]. Moreover, characterizing the burden of RSV-related disease in adults depends on obtaining accurate statistics on the incidence, hospitalization rate, and hospital mortality rates of RSV infected adults in developing nations. Though influenza has traditionally received more attention due to its higher prevalence and the attendant ease of diagnosis, there is mounting evidence that RSV infection is actually associated with worse outcomes and greater medical resource utilization than influenza infected patients [3,5,22]. As recognition of RSV-associated illness continues to grow, it is likely that our understanding of its importance will as well.

## Data Availability

Not applicable.

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
