# Peer review of "Going Viral—RSV as the Neglected Adult Respiratory Virus"

_pathogens, 2022, doi:10.3390/pathogens11111324_

Round 1

Reviewer 1 Report

Busack and Shor wrote a briliant review about infection by Respiratory Syncytial Virus. This virus is responsible for acute respiratory infections, and is described mainly in infants with bronchiolitis and in the ederly. RSV is not often studied in adulthood and the burden of its infection is still unclear. The manuscript is clear, well organized and easily understandable. It brings an adjustment. It provides an update on the diagnosis, risk factors for severity, and the development and research of vaccines.

Author Response

Response to Reviewers

Reviewer 1.

We appreciate the reviewer’s kind words regarding our paper.  S/he made no specific requests for revisions.

Reviewer 2.

We again appreciate the reviewer’s positive comments regarding the manuscript.

  1. We have replaced the first two paragraphs of section 6 with the recommended language.

Reviewer 3.

  1. We have revised the title as suggested by the reviewer – we appreciate his important observation.
  2. We concur that the paragraph is unclear and have deleted the first sentence and clarified the remainder of the text. This makes the distinction between the adult vaccines and the monoclonals clearer.
  3. We have clarified the text to address the important points the reviewer notes.
  4. We thank the reviewer for catching our oversight re the J & J and BN vaccines. We have added a sentence now stating this.
  5. We respectfully disagree with the reviewer regarding the value of an illustration given the space limitations and other factors.
  6. We have gone through the manuscript and believe all the references are used correctly and ordered correctly.

Reviewer 2 Report

This paper is an excellent review of the importance of RSV in elders. It highlights new analyses of patient samples that indicate a heavier burden that influenza virus. It will be a useful review for investigators and clinicians, alike.

I have made some comments on the manuscript that will improve the article, particularly in the section describing the virus and its classification. 

Please find my modification for page 5 below:

RSV is a non-segmented negative sense RNA virus in the family Pneumoviridae, genus Orthopneumoviridae. Its pleomorphic, enveloped virion ranges from spherical to rod-shaped, 120-300 nanometers in diameter [1]. RSV has two major antigenic subgroups (A and B). Although these two subgroups have antigenic differences in a number of proteins, the major genetic diversity lies within G, the attachment glycoprotein [1,4,7]. Once attached, the F (fusion) protein undergoes structural changes that cause the virion membrane to fuse with the cell membrane, initiating infection. When the F protein produced in infected cell lines reaches the surface, it causes infected cells to fuse with adjacent uninfected cells, forming “syncytia” or giant cells [4]. For several reasons the fusion protein has become paramount in the development of RSV vaccines. It is relatively conserved among strains, such that a vaccine utilizing a recombinant form of the RSV-F protein would likely be effective against both RSV A and B subgroup viruses. In addition, the “prefusion” form of the F protein has an antigenic site (theta) that induces very potent RSV neutralizing antibodies [4,26].

              Currently, one passive vaccine is available, palivizumab (Synergis), licensed for use in high-risk infants [26]. It prevents roughly 50% of RSV related hospitalizations in premature infants, however, it is costly and must be administered monthly over the infant’s first 5-month RSV season. Palivizumab is a humanized monoclonal antibody directed against the “postfusion” form of the RSV F protein. Other monoclonal antibodies are currently in development that recognize the prefusion F protein and therefore are more effective at neutralizing RSV. These antibodies are also stabilized so that multiple doses are not necessary.

Author Response

(The authors gave the same response as above.)

Reviewer 3 Report

Pathogens-2000405 

According to its abstract, this review manuscript is focused on the importance of RSV infection in adults. However, the work needs some modifications to reflect that focus. Specific comments are as follows:

1.     The title needs to reflect that the review is focused on adult infection, otherwise is misleading. Besides, that change would distinguish this work from recent RSV reviews. 

2.     Page 5, line 199. The authors start the paragraph indicating the licensed RSV vaccine. Then, they mention the monoclonal antibody. Then they refer again to the vaccine “the vaccine is effective….” Which vaccine? The one that is licensed? Why mention the palivizumab in between the vaccine information? That paragraph needs to be organized so that there is no ambiguity in the idea they are trying to convey.  

3.     A similar observation in page 6, line 208. The authors started to talk about vaccines, then 2  MAb, “one developed by AstraZeneca…The other is MK-1654 (MERK)”. Then they continue with: “Unfortunately, both vaccines are currently being investigated…” which vaccines are they referring to? That paragraph needs to be rewritten; it is confusing for the reader.

4.      The information on the development of vaccines is incomplete. The Janssen Pharmaceutical and the Bavarian Nordic are also in Phase 3. That needs to be discussed concerning their efficacy. 

5.     An illustration is needed to be included to summarize one of the main aspects of this review. 

6.     Finally, the numbers in the “references” section need to be revised, specifically, the one for reference 3, as it appears as 3. then 1. 

Author Response

(The authors gave the same response as above.)
